# Improving Foodborne Pathogen Control Using Green Nanosized Emulsions of *Plectranthus hadiensis* Phytochemicals

**Lucía Carolina Vega-Hernández [1], Julio César Serrano-Niño [1], Carlos Arnulfo Velázquez-Carriles [1,2], Alma H. Martínez-Preciado [3], Adriana Cavazos-Garduño [1,\*] and Jorge Manuel Silva-Jara [1,\*]**

[1] Departamento de Farmacobiología, Centro Universitario de Ciencias Exactas e Ingenierías, Universidad de Guadalajara, Blvd. Marcelino García Barragán 1421, Guadalajara 44430, Jalisco, Mexico; lucia.vega1159@alumnos.udg.mx (L.C.V.-H.); julio.serrano@academicos.udg.mx (J.C.S.-N.); arnulfo.velazquez@academicos.udg.mx (C.A.V.-C.)

[2] Departamento de Ingeniería Biológica, Sintética y de Materiales, Centro Universitario de Tlajomulco (CUTLAJO), Universidad de Guadalajara, Carretera Tlajomulco, Santa Fé, Km 3.5, 595, Tlajomulco de Zúñiga 45641, Jalisco, Mexico

[3] Departamento de Ingeniería Química, Centro Universitario de Ciencias Exactas e Ingenierías, Universidad de Guadalajara, Blvd. Marcelino García Barragán 1421, Guadalajara 44430, Jalisco, Mexico; alma.martinez@academicos.udg.mx

\* Correspondence: adriana.cavazos@academicos.udg.mx (A.C.-G.); jorge.silva@academicos.udg.mx (J.M.S.-J.)

**Abstract:** Every year, millions of foodborne illnesses with thousands of deaths occur worldwide, which is why controlling foodborne pathogens is sought. In this study, nanoemulsions of phytochemicals extracted from *Plectranthus hadiensis* var. *tomentosus* (PHT) were obtained, and their antioxidant and antimicrobial capacities were evaluated. PHT extracts were obtained by maceration, ultrasound, and Naviglio methods, and their antimicrobial activity against *Staphylococcus aureus*, *Listeria monocytogenes*, *Escherichia coli*, and *Salmonella enterica* was determined by the microdilution method. The extract with the highest antimicrobial activity was obtained by Naviglio with a minimum inhibitory concentration (MIC) and minimum bactericidal concentration (MBC) of 12.5 and 25 mg/mL, respectively, for all bacterial strains. The nanoemulsion (o/w) made with Tween 40, 5% extract, and 50% ultrasonic amplitude had a globule size of 4.4 nm, a polydispersity index of 0.48, and a surface charge of −0.08 mV and remained stable for 30 days. This nanosystem presented significantly higher antimicrobial and antioxidant activity than the free extract. Thus, the nanoencapsulation of the phytochemicals in the PHT extracts is an alternative to protect and enhance their biological activity against pathogenic microorganisms.

**Keywords:** *P. hadiensis*; nanoemulsions; antioxidant capacity; antibacterial activity

## 1. Introduction

Foodborne diseases are one of the leading public health problems and concern consumers, the food industry, and regulatory authorities [1]. The most common symptoms of foodborne diseases are abdominal pain, diarrhea, vomiting, nausea, fever, and, in severe cases, death. The most important pathogenic bacteria causing foodborne illnesses are *Campylobacter*, *Salmonella*, *Listeria monocytogenes*, pathogenic *Escherichia coli*, *Staphylococcus aureus*, *Clostridium perfringens*, *Bacillus cereus*, and *Yersinia enterocolitica* and have been associated with foods such as red and poultry meats, fish and shellfish, milk and dairy products, and fruits and vegetables [2].

Numerous species of the genus *Plectranthus* are used in traditional medicine to treat ailments due to their content of bioactive compounds, such as diterpenoids and polyphenols that exhibit antibacterial and antifungal properties [3–5]. In the case of *Plectranthus hadiensis* var. *tomentosus* (PHT) or "*VapoRub*TM *plant*", as it is commonly called in Mexico, it has been used to treat various digestive, respiratory, and skin problems, such as dysentery,

vomiting, nausea, diarrhea, fever, sore throat, common cold, asthma, inflammations, and smallpox, because it is a rich source of bioactive phytochemicals, especially terpenoids. The juice from the leaves and stems mixed with honey treats diarrhea, while its infusions treat cough [6–10].

Bioactive compounds can be extracted from plant sources through conventional and non-conventional methods. Conventional methods are the most widely used and consist of placing the plant matrix in contact with the solvent under agitation, where bioactive compounds are extracted due to diffusion and mass transfer phenomena [11]. These methods include maceration, decoction, percolation, hydro-distillation, and Soxhlet extraction [12]. However, these methods have certain limitations, as they require large amounts of solvents, have low selectivity in extraction, are characterized by long extraction times, and can cause the degradation of thermolabile compounds [13]. On the other hand, non-conventional methods are more environmentally friendly due to the reduced use of solvents, the shorter operation time, and the production of better-quality extracts with higher yield. These methods include ultrasound-assisted extraction, supercritical fluid extraction, microwave-assisted extraction, accelerated solvent extraction, enzyme-assisted extraction, and dynamic solid–liquid extraction [12]. The extraction method plays an essential role in the biological activities that bioactive compounds may present, such as antimicrobial and antioxidant effects, as chemical solvents or heat can affect their functionality [14].

The Naviglio method is an innovative technology that works at room temperature through pressurization–depressurization cycles, facilitating the extraction of bioactive compounds faster than conventional extraction technologies. It is used in an aqueous or alcoholic medium and is favored in extracting thermolabile phenolic compounds with antimicrobial activity [15,16]. The extraction cycle consists of a static and a dynamic phase. During the static phase, the system is subjected to pressure for a specific time during which the liquid penetrates the solid and permeates through the cell membrane. At the end of this stage, the dynamic phase immediately begins, in which the pressure returns to its initial condition, generating a negative pressure gradient between the inside and outside of the solid matrix, resulting in a suction effect that extracts the phenolic compounds to the solvent [15]. Unlike other solid–liquid extractions, this method is an "active" process because the pressure gradient forces the exit of compounds. In contrast, other techniques based on diffusion and osmosis are "passive" processes [13].

Despite the in vitro antimicrobial and antioxidant efficacy of extracts through unconventional extraction techniques, these present certain limitations, such as their low solubility in aqueous systems, chemical instability against factors such as light, heat, moisture, and oxygen, interactions with the components of the food matrix, and inactivation by enzymatic degradation, making their direct use in food limited [17–19]. In addition, the intense aroma of most natural antimicrobials, even at low concentrations, can negatively influence the sensory characteristics of the food, leading to a decrease in consumer acceptance [17].

Among the different types of nanosystems, nanoemulsions are the most used for nanoencapsulation of bioactive compounds [20]. Nanoemulsions consist of the dispersion of two immiscible liquid phases, where one is dispersed in the other in the form of globules that are on a nanometric scale, that is, between 20 and 500 nm, and are composed of an aqueous phase, an oily phase, and an emulsifier [21,22]. These colloidal systems can encapsulate bioactive compounds in significant quantities by trapping them within the dispersed phase stabilized by surfactants or biopolymers in the continuous phase. Among the most used surfactants are phospholipids, proteins, low-molecular-weight polysaccharides, and short-chain surfactants such as Tween and Span. Nanoemulsions are characterized by being transparent or translucent in appearance and having small sizes between 20 and 200 nm. They also have high kinetic stability and can easily incorporate phytochemical compounds of both hydrophilic and lipophilic nature [23]. In turn, the development of nanoemulsions by ultrasound has gained popularity due to its low cost, as its primary function is to generate acoustic cavitations forming microbubbles that collapse in the system, generating turbulence that is responsible for the reduction in particle size and, therefore,

allows for a reduction in the amount of surfactant to be used. There are reports where the in vitro antimicrobial activity of vegetable extract nanoemulsions such as oregano or thyme has been evaluated and compared, and it has been shown that their bactericidal activity significantly increases compared with the free extract [24].

In this work, the phytochemical profile of PHT ethanolic extracts obtained by maceration, ultrasound, and Naviglio was studied in the first stage. In the second stage, the best extract with antibacterial activity (the Naviglio extract) was selected, and the effect of nanoencapsulation by ultrasound using Tween 40 as a surfactant was studied, in which narrow-distribution particle sizes were obtained, thereby enhancing the antioxidant and antimicrobial activity in vitro against foodborne pathogens compared with the free or unencapsulated extract. In addition, the stability of the nanoemulsion at room temperature was evaluated.

## 2. Materials and Methods

### 2.1. Plant Material

*Plectranthus hadiensis* var. *tomentosus* (PHT) plants were obtained from local markets in Guadalajara, Mexico. The aerial parts were washed and disinfected by immersion in a sodium hypochlorite solution at 80 ppm. Afterward, they were dried at 50 °C for 48 h in a dehydrator (Hamilton Beach 32100A, Richmond, VA, USA) and powdered in a blade processor.

### 2.2. Extraction of Plant Material

Plant extracts were obtained by Naviglo's technique using an Armfield (FT 110, Hampshire, UK) extractor by placing 100 g of the powdered aerial parts with 1.3 L of 96% ethanol in the extraction chamber (filter bag, Ø 5 μm). The extraction conditions consisted of 15 cycles of 10 min (5 min of static phase and 5 min of dynamic phase) with a maximum pressure of 7 bar. In order to compare the antimicrobial activity, extracts were also obtained by maceration and ultrasound. Extraction by maceration was performed by placing 25 g of the powdered aerial parts with 250 mL of ethanol in agitation for 48 h. In contrast, the ultrasound-assisted extraction was performed using a sonicator (Branson SFX 550, Emerson Electric Co., St. Louis, MO, USA) with 130 J of energy and an amplitude of 20%. The extracts were concentrated at 50 °C in a rotary evaporator (Buchi R-100, New Castle, DE, USA) and stored at −15 °C until further analysis.

### 2.3. Extract Characterization

The extract was analyzed in a Fourier transform infrared (FTIR) spectrophotometer (Cary 630, Agilent Technologies, Santa Clara, CA, USA). The extract was positioned in the diamond crystal, and spectra were recorded at 4000 to 500 cm$^{-1}$ at a nominal resolution of 4 cm$^{-1}$ and acquired at 32 scans per sample.

### 2.4. Phenolic Compound Quantification and Antioxidant Activity

To quantify the phenolic compounds and antioxidant capacity, extracts were dissolved with 96% ethanol to a 1 mg/mL concentration for both methods.

#### 2.4.1. Phenolic Compound Quantification

The total phenolic content was evaluated using the Folin–Ciocalteu colorimetric method [25] with gallic acid as a reference standard. A total of 250 μL of the extract, 1500 μL of distilled water, and 125 μL of the Folin–Ciocalteu reagent (F9252, Sigma-Aldrich, Darmstadt, Germany) were placed in a tube and left to react for 5 min. Subsequently, 500 μL of distilled water and 375 μL of 7.5% $Na_2CO_3$ (26240, Golden Bell, Villa Milpa Alta, México) solution were added, homogenized, and incubated for 2 h in the dark. After incubation, the absorbance was read at 750 nm in a spectrophotometer (Multiskan SkyHigh, Thermo Scientific, Waltham, MA, USA). The total phenolic content is expressed as milligrams of

gallic acid equivalents (GAE)/g extract. The standard curve was used to calculate GAE/g extract (y = 0.004 − 0.0359, $r^2$ = 0.998).

### 2.4.2. Total Flavonoid Content

The methodology employed was based on that proposed by Panche et al. [26]. In a test tube, 250 μL of the extract, 1 mL of distilled water, and 75 μL of 5% NaNO (Sigma-Aldrich) were placed, and it was incubated at 25 ± 5 °C for 5 min. Subsequently, 75 μL of 10% $AlCl_3$ (Sigma-Aldrich) was added, and it was incubated at 25 ± 5 °C for 6 min. Finally, 600 μL of distilled water and 500 μL of 1 M NaOH (Sigma-Aldrich) were added, homogenized, and 200 μL of each reaction was placed in triplicate on microplates. The absorbance at 510 nm was then measured. The standard curve was used to calculate quercetin equivalents QE/g extract (y = 0.0003x + 0.0223, $r^2$ = 0.996).

### 2.4.3. DPPH Radical Scavenging Activity

The scavenging effect of the extracts on the radical 2,2-diphenylpicrylhydrazyl (DPPH) was evaluated employing the colorimetric method described by Brand-Williams, Cuvelier, and Berset [27]. Briefly, 200 μL of the sample, 1350 μL of DPPH (D9132, Sigma-Aldrich) solution, and 150 μM absolute methanol were incubated for 30 min in the dark at the ambient temperature. After incubation, the absorbance was read at 517 nm. The radical scavenging activity is expressed as a percentage and was calculated from the equation:

$$DPPH \; scavenging \; activity \; (\%) = \frac{\left(Abs_{control} - Abs_{sample}\right)}{Abs_{control}} \times 100 \tag{1}$$

where $ABS_{sample}$ = the absorbance of the test sample, while $ABS_{control}$ = the absorbance of the negative control (methanol).

### 2.4.4. Ferric-Reducing Antioxidant Power Assay

The ferric-reducing activity of the extracts was evaluated by the reducing ability of plasma (FRAP) assay developed by Benzie and Strain [28]. The FRAP reagent consisted of 6 mL of 10 mM 2,4,6-Tris (2-pyridyl)-s-triazine (TPTZ, T1253, Sigma-Aldrich) solution in 40 mM HCl, 6 mL of 20 mM $FeCl_3$ (157740, Sigma-Aldrich), and 60 mL of 300 mM acetate buffer (pH 3.6). A total of 75 μL of the extract was added to 1425 μL of FRAP reagent and incubated in the dark for 30 min at 37 °C. The absorbance was read at 593 nm, and the antioxidant activity is expressed as milli Molar of Trolox equivalents (mM TE)/g extract. The standard curve was used to calculate Trolox equivalents TE/g extract (y = 0.0014x + 0.0008, $r^2$ = 0.994).

### 2.5. Antibacterial Activity

#### 2.5.1. Microorganisms and Growth Conditions

The antibacterial activity of the extracts was evaluated against four bacterial species: *Escherichia coli* ATCC 8739, *Staphylococcus aureus* ATCC 25923, *Salmonella enterica* isolated from mangoes by the Universidad Autónoma de Querétaro, and *Listeria monocytogenes* isolated from cheese and sequenced and donated by the Molecular Biology Laboratory of the Universidad de Guadalajara. The strains were activated in trypticase soy broth (7100A, Neogen, Lansing, MI, USA) and incubated at 37 °C for 18 h. After incubation, the fresh culture was adjusted to $1.5 \times 10^8$ CFU/mL using the 0.5 MacFarland turbidity scale and diluted with sterile saline solution to achieve a $5 \times 10^6$ CFU/mL concentration.

#### 2.5.2. Minimum Inhibitory Concentration (MIC)

The minimum inhibitory concentration was evaluated using the broth microdilution method described in [4], incorporating resazurin as a colorimetric indicator of cell growth. The extracts were dissolved with dimethyl sulfoxide DMSO and Mueller–Hinton broth (70192, Sigma-Aldrich, Darmstadt, Germany) at a 200 mg/mL concentration and filtered

using sterile syringe filters of 0.45 µm. In aseptic conditions, 100 µL of Mueller–Hinton broth (MHB) was added to each well of a 96-well plate. To the first well of each column was added 100 µL of the extract, and a 1:2 serial dilution was made to achieve concentrations between 100 and 1.56 mg/mL using a multichannel micropipette. Then, 10 µL of the inoculum was added to each well, resulting in a final bacterial concentration of $5 \times 10^5$ CFU/mL in the wells. A positive control of ampicillin (100 µg/mL) and a negative control of MHB and DMSO were used. The plates were incubated at 37 °C for 24 h. After incubation, 30 µL of 0.015% resazurin (199303, Sigma-Aldrich) was added to all the wells and further incubated for 1 h to observe color changes. Any color change from blue to pink was recorded as bacterial growth, while blue was interpreted as growth inhibition. The lowest concentration at which no color change occurred was taken as the MIC value. Assays were carried out in triplicate for each tested microorganism.

### 2.5.3. Minimum Bactericidal Concentration (MBC)

The minimum bactericidal concentration was determined according to Jugreet and Mahomoodally [29]. Briefly, 10 µL of the well content with no color change from the MIC assay was inoculated on Mueller–Hinton agar (21667, BD Bioxon, Nuevo León, Mexico) plates and incubated at 37 °C for 24 h. The MBC value was determined as the lowest concentration with no visible microbial growth on the plates. Positive and negative controls were used, and the assays were carried out in triplicate.

### 2.6. Acute Toxicity Assay

Brine shrimp (*Artemia salina*) eggs were hatched in synthetic seawater (24 g NaCl/L) at 28 °C under light and continuous aeration. Then, 10 nauplii were transferred per well with 5 mL of synthetic seawater in 6-well microplates and 7 mm diameter Whatman No. 4 disks, on which Naviglio extract treatments at different concentrations (20–100 mg/mL) had previously been deposited and evaporated. Silver sulfate was used as the positive control and synthetic seawater was used as the negative control. After 24 h, the nauplii were counted, and the $LD_{50}$ was determined using the PROBIT method. The toxicity was classified according to the $LD_{50}$ value (mg/mL): non-toxic ($LD_{50} > 1000$), slightly toxic ($500 < LD_{50} < 1000$), moderately toxic ($100 < LD_{50} < 500$), and extremely toxic ($LD_{50} < 100$).

### 2.7. Preparation of Ultrasound-Assisted Nanoemulsions

PHT nanoemulsions were prepared using PHT extract as the dispersed phase, deionized water as the continuous phase, and Tween 40 (15% *w/w*) as the emulsifier. First, coarse emulsions were prepared by high-speed stirring using a homogenizer (Ultra-Turrax T25, IKA, Wilmington, NC, USA) at 8000 rpm for 20 min. After that, the obtained coarse emulsions were subjected to ultrasonic emulsification for 20 min in a Branson sonicator (SFX 550, Emerson Electric Co., St. Louis, MO, USA) with a power of 550 W and a frequency of 20 kHz. The heat generated during sonication was eliminated using an iced water bath. A set of nanoemulsion formulations were prepared to evaluate the effect of two variables using a $2^2$ factorial design. The studied variables were extract concentration (5 and 10% *w/w*) and ultrasonication amplitude (40% and 50%), as shown in Table 1. The nanoemulsions were characterized, and the storage stability was evaluated.

**Table 1.** Treatments of the factorial design for the preparation of the nanoemulsions.

| Treatment | Extract Concentration (%) | Amplitude (%) |
|:---:|:---:|:---:|
| NE1 | 5 | 40 |
| NE2 | 5 | 50 |
| NE3 | 10 | 40 |
| NE4 | 10 | 50 |

### 2.8. Nanoemulsion Characterization

Characterization of the nanoemulsions was described by the average droplet size (Z-average), polydispersity index (PDI), and zeta potential. These properties were determined in a Zetasizer Nano S90 (Malvern Instruments Inc., Worcestershire, UK), and all the measurements were made in triplicate at 25 °C.

### 2.9. Storage Stability Study

The storage stability of nanoemulsions was evaluated by measuring the change in the droplet size and PDI for 30 days at the ambient temperature. Nanoemulsions considered stable were those without phase separation and droplet sizes below 200 nm [30]. In addition, the total phenolic content and antioxidant capacity were evaluated during the storage of nanoemulsions.

### 2.10. Encapsulation Efficiency (EE)

The encapsulation efficiency of the most stable nanoemulsion was evaluated using the methodology reported by Jan et al. [31] with some modifications. A total of 1 mL of the nanoemulsion was placed in duplicate in a centrifuge tube and centrifuged at 10,000 rpm for 15 min. The encapsulation efficiency was determined by measuring the total phenolic content of the nanoemulsion according to the following formula:

$$EE(\%) = \frac{PT_f}{PT_i} \times 100 \tag{2}$$

where $PT_i$ and $PT_f$ are the total phenolic content of the nanoemulsion before and after centrifugation, respectively.

### 2.11. In Vitro Release Study

An in vitro release study of the nanoemulsion and crude extract was performed using a dialysis membrane (Fisherbrand regenerated cellulose, pore size: 12–14 kDa) according to the methodology described by Pathania et al. [32]. The membranes were pretreated by immersing them in distilled water for 48 h. The nanoemulsion or extract (3 mL) was loaded into the dialysis membrane and placed in beakers with 200 mL of phosphate buffer solution (pH 7.4) in a shaking incubator for 48 h at 37 °C and 75 rpm. At different time intervals, 5 mL of the sample was taken and replaced with the same volume of fresh PBS. The assay was duplicated, and the total polyphenol content was evaluated in the different samples to obtain a release curve.

### 2.12. Phenolic Compound Quantification and Antioxidant Activity of the Nanoemulsion

The phenolic compound quantification and antioxidant capacity of the PHT nanoemulsion and crude extract were determined as described previously in Section 2.4. For both methods, the crude extract was dissolved with water and Tween 40 at the same concentration as the nanoemulsion and homogenized in a vortex for 30 s. Then, both the nanoemulsion and the extract were diluted with water to a concentration of 1 mg/mL.

### 2.13. Antibacterial Activity of the Nanoemulsion

To compare the antimicrobial activity of the nanoemulsion with that of the crude extract, the MIC and MBC were determined as described previously. The extract was dissolved with water and Tween 40 at the same concentration as the nanoemulsion, and both the extract and nanoemulsion were filtered using sterile syringe filters of 0.45 μm. Serial dilutions were made to obtain 25 to 0.39 mg/mL concentrations. A positive control of ampicillin (100 μg/mL) and a negative control of deionized water and Tween (15% *w/w*) were used.

### 2.14. Effect of the Extract and Nanoemulsion on the Growth of Bacteria

In order to evaluate the interaction between the nanoemulsion or the extract with a Gram-negative bacterium (*E. coli*) and a Gram-positive bacterium (*S. aureus*), their effect on the growth of the two bacteria was studied according to the methodology of Qi et al. [33] with some modifications. For this, 5 mL of sterile Mueller–Hinton broth, 5 mL of the nanoemulsion or extract, and 50 μL of the inoculum adjusted to 0.5 McFarland were placed in sterile tubes. In contrast, 5 mL of Mueller–Hinton broth, 5 mL of sterile distilled water, and the inoculum were placed for growth control. The tubes were incubated at 37 °C for 48 h, and their optical density was read at 600 nm using a spectrophotometer (Metash UV-5100, Shanghai Metash Instruments Co.,Ltd., Shanghai, China).

### 2.15. Statistical Analysis

The measurements were carried out in triplicate, and the results are expressed as the mean value $\pm$ standard deviation. Comparisons were performed using one-way ANOVA followed by the LSD test ($p < 0.05$) using the statistical software Statgraphics19® (V-19.5.01).

## 3. Results

### 3.1. Extract Characterization

The IR spectra (Figure 1a) of PHT extracts show the same signals, highlighting functional groups such as phenols (3336 cm$^{-1}$), alkanes (2924 cm$^{-1}$), aldehydes (2853 cm$^{-1}$), aromatic compounds (1457 cm$^{-1}$), and carboxylic acids (1043 cm$^{-1}$). The broad band at ca. 3336 cm$^{-1}$ is linked to phenolic compounds for the genus *Plectranthus* [34]. At the same time, clear signals can be seen for the family of terpenoids and monoterpenes attributable to the characteristic odor of the plant at 1353, 880, and 814 cm$^{-1}$ [35,36]. To determine differences between the extraction methods, a second derivative was applied (Figure 1b), in which the absorbance was more intense at 1738 cm$^{-1}$ for the Naviglio extract, denoting –C=O ester groups. Also, the band at 1162 cm$^{-1}$ could be related to C–C–H vibrations corresponding to dihydroxyphenyl aromatic rings. Finally, at around 695 cm$^{-1}$, C–H monosubstituted benzoic ring vibrations are associated [37].

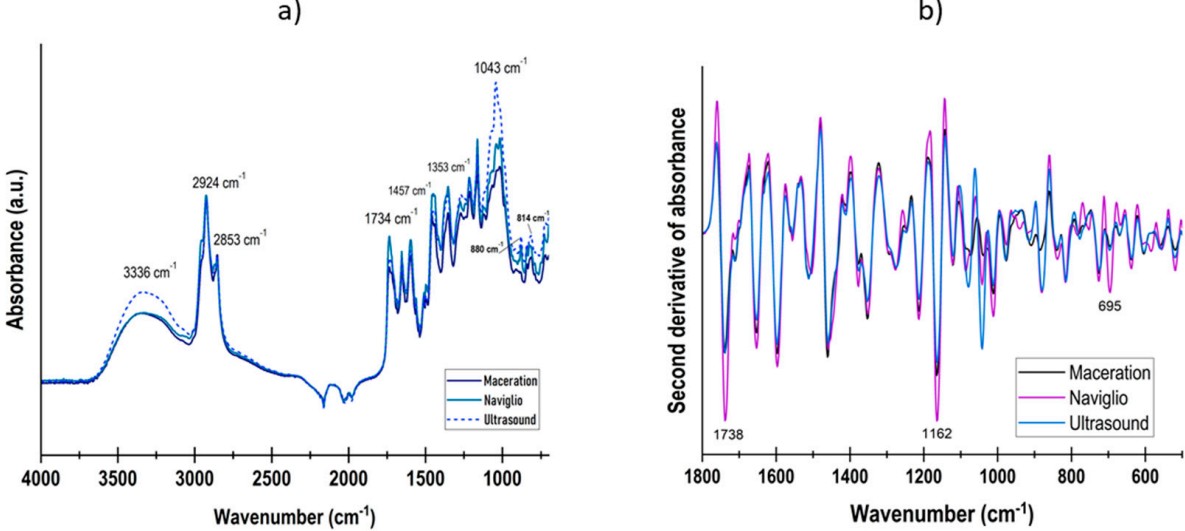

**Figure 1.** FTIR spectra of (**a**) *Plectranthus hadiensis* extracts and (**b**) the second derivative in the region from 1800 cm$^{-1}$ to 500 cm$^{-1}$.

The total polyphenol content of the extracts ranged from 95.49 to 104.9 mg GAE/g (Table 2). A significant difference ($p < 0.05$) was observed between the extracts obtained by Naviglio and maceration. In the DPPH radical inhibition test, the extracts showed an antioxidant capacity of 495 $\pm$ 69 mM Trolox/g for the extract obtained by ultrasound, 258 $\pm$ 75 mM Trolox/g for Naviglio, and 296 $\pm$ 76 mM Trolox/g for maceration, with DPPH

radical inhibition percentages of 42, 21, and 25%, respectively. Although in this study a phenolic profile was not searched for, the antioxidant activity exhibited for each extraction method is related to compounds such as rosmarinic acid, casticin, and ayanin, which are present in *P. hadiensis* extracts as was found by Ji et al. [3] through HPLC. These results are similar to those obtained by Rijo et al. [4], who reported an inhibition percentage between 18 and 40% for aqueous PHT extracts obtained by decoction, infusion, and microwaves. Moreover, the results also coincide with those of Ntungwe et al. [38], who reported an inhibition percentage of 36% for the ultrasound-obtained extract of *P. hadiensis*. For the ferric-reducing antioxidant power test, an antioxidant capacity of $666 \pm 49$ mM Trolox/g was obtained for the extract obtained by ultrasound, $360 \pm 23$ mM Trolox/g for the extract obtained by Naviglio, and $527 \pm 23$ mM Trolox/g for the extract obtained by maceration. These results are higher than those obtained by Muniyandi et al. [39], who reported an antioxidant capacity of 154 mg EFe(II)/g for the methanolic extract of *Plectranthus stocksii* obtained by Soxhlet, possibly because this method can degrade thermolabile antioxidant compounds due to the temperature used. It has been demonstrated that when Soxhlet extraction is applied for a prolonged time, the antioxidant activity of the extract decreases, as has been shown for thyme, hemp, and coriander [40].

**Table 2.** Polyphenols and antioxidant activity contents by different extraction methods.

| Extraction Method | Polyphenols [1] | Flavonoids [2] | Antioxidant Capacity | |
| --- | --- | --- | --- | --- |
| | | | DPPH [3] | FRAP [4] |
| Maceration | $95.49 \pm 5.3$ [a] | $5.77 \pm 1.02$ [a] | $295.72 \pm 75.8$ [a] | $527.31 \pm 23.1$ [a] |
| Naviglio | $104.9 \pm 6.8$ [b] | $4.57 \pm 0.63$ [b] | $258.03 \pm 74.8$ [a] | $359.64 \pm 23.4$ [b] |
| Ultrasound | $97.58 \pm 9.0$ [ab] | $5.98 \pm 0.79$ [a] | $494.96 \pm 68.9$ [b] | $665.81 \pm 49.1$ [c] |

Different letters by column indicate significant differences ($p < 0.05$). [1] Expressed as mg of gallic acid equivalents (GAEs) per g of dry extract. [2] Expressed as mg of quercetin equivalents (QEs) per g of dry extract. [3,4] Expressed as mM of Trolox equivalents per g of dry extract.

The extraction method can influence the content of phenolic compounds in the extracts due to extraction conditions such as temperature, pressure, time, and type of solvent used [12,18]. Ji et al. [3] reported 130 and 48 mg CE/g in PHT extracts obtained by maceration from stems and leaves, respectively, while [41] obtained 80 mg GAE/g in PHT methanolic extracts by Soxhlet extraction. In addition, Schultz et al. [42] reported 18 and 8 mg caffeic acid equivalent CAE/g in extracts obtained by maceration and Soxhlet, respectively. Regarding extraction methods, both in the DPPH and FRAP tests, the extract obtained by ultrasound showed a significantly higher antioxidant capacity ($p < 0.05$), followed by the extract obtained by maceration and the one obtained by Naviglio. These results are in line with what was reported by Rocchetti et al. [11], who obtained a higher antioxidant capacity evaluated by the FRAP method in the *M. oleifera* extract obtained by ultrasound, followed by the one obtained by maceration and, finally, the one obtained by Naviglio, which showed the lowest antioxidant capacity. The authors observed that the extraction method influenced the type of phenolic compound obtained in the extracts, especially phenolic acids and flavonoids. When relating the type of phenolic compound extracted with antioxidant capacity, they observed that it was influenced by the concentration of flavonols, flavones, and anthocyanins, which belong to the flavonoid family. The extract obtained by ultrasound was rich in phenolic acids and flavonoids such as flavones, flavonols, and anthocyanins, which are strong antioxidants. In contrast, the extract obtained by Naviglio had a higher concentration of hydroxycoumarins, stilbenes, and other polyphenols. Similarly, Sánchez-Gómez et al. [43] compared the polyphenol content and antioxidant capacity of vine shoot extracts obtained by decoction, microwaves, and Naviglio. They reported that the extract obtained by Naviglio had a lower antioxidant capacity because it had a concentration of flavonols and phenolic acids that was 3 to 6 times lower than the other extracts. In this study, flavonoids were quantified to verify these assumptions, where lower concentrations were found compared with maceration and ultra-

sound ($p < 0.05$). These findings coincide with what was obtained in this study. Although the extract obtained by Naviglio had a polyphenol content similar to that obtained by ultrasound, the antioxidant capacity was significantly different, which can be attributed to the type of phenolic compound extracted with the different extraction methods.

### 3.2. Antimicrobial Activity of the Extracts

All PHT extracts showed antimicrobial activity against both Gram-positive and Gram-negative bacteria, which differs from previous reports [10,37,44,45] where PHT extracts exhibited antimicrobial activity only against Gram-positive bacteria such as *Staphylococcus aureus*, *Bacillus subtilis*, *Micrococcus flavus*, and *Enterococcus faecalis*. However, Kotagiri, Shaik, and Kolluru [8], as well as Sripathi and Ravi [46], reported that in addition to having antimicrobial activity against Gram-positive bacteria, PHT extracts also showed activity against Gram-negative bacteria such as *Escherichia coli* and *Pseudomonas aeruginosa*. According to Sripathi, Jayagopal, and Ravi [47], the antimicrobial activity of *P. hadiensis* can be attributed to the presence of monoterpenes and sesquiterpenes, mainly L-fenchone, β-farnesene, copaene, and α-caryophyllene, while Dominguez-Martín et al. [10] and Napagoda et al. [48] ascribe the antimicrobial activity mainly to the diterpenes 7α-acetoxy-6β-hydroxyroyleanone, 7 β-acetoxy-6β-hydroxyroyleanone, and 6β,7β-dihydroxyroyleanone.

The MIC and MBC values of the different extracts are shown in Table 3. The MIC value of the extracts ranged between 12.5 and 25 mg/mL, while the MBC ranged between 12.5 and 50 mg/mL. These results differ from those obtained by other authors. Sripathi, Jayagopal, and Ravi [47] reported an MIC value of 320 μg/mL for *S. aureus* and *E. coli* in extracts obtained by hydrodistillation. In contrast, the methanolic extracts obtained by Napagoda et al. [48] showed an MIC of 500 μg/mL for *S. aureus* and 1000 μg/mL for *Salmonella* typhi. In addition, Ntungwe et al. [38] reported an MIC and MBC value of 15.6 μg/mL and 250 μg/mL, respectively, for *S. aureus* in the ultrasound-assisted extract and Schultz et al. [42] obtained an MIC of 104 μg/mL for *S. aureus* and an MIC of >500 μg/mL for *E. coli* and *Listeria innocua*. The difference in the antimicrobial activity of the extracts can be due to the extraction method and solvent used. Also, factors such as geographic location, environment, growing conditions, growing season, harvest time, and age of the plant can affect the content of monoterpenes, sesquiterpenes, and diterpenes present in the extracts and, hence, their antimicrobial activity [49–51]. Sripathi, Jayagopal, and Ravi [47] observed a lower MIC value in extracts obtained during the summer season (320 μg/mL) than in the one obtained during the rainy season (640 μg/mL).

**Table 3.** The minimum inhibitory concentration (MIC) and minimum bactericidal concentration (MBC) of *Plectranthus hadiensis* extracts obtained by different extraction methods.

| Bacteria | MIC (mg/mL) [1] | | | MBC (mg/mL) [1] | | |
|---|---|---|---|---|---|---|
| | Maceration | Ultrasound | Naviglio | Maceration | Ultrasound | Naviglio |
| *Staphylococcus aureus* | 25 [a] | 25 [a] | 12.5 [b] | 50 [a] | 50 [a] | 25 [b] |
| *Listeria monocytogenes* | 25 [a] | 25 [a] | 12.5 [b] | 25 [a] | 25 [a] | 12.5 [b] |
| *Escherichia coli* | 25 [a] | 25 [a] | 12.5 [b] | 25 [a] | 25 [a] | 25 [b] |
| *Salmonella enterica* | 25 [a] | 25 [a] | 12.5 [b] | 25 [a] | 25 [a] | 25 [b] |

[1] Values represent the average of three measurements. Within rows, values with different letters are significantly different ($p < 0.05$).

When comparing the different extracts, those obtained by maceration and ultrasound presented an MIC value of 25 mg/mL. At the same time, the one obtained by Naviglio had a lower MIC of 12.5 mg/mL. Regarding the MBC, in Gram-negative bacteria, there was no significant difference ($p > 0.05$) in the MBC value for the extraction methods. In contrast,

in Gram-positive bacteria, the extract obtained by Naviglio showed a lower MBC with 25 and 12.5 mg/mL for *S. aureus* and *L. monocytogenes*, respectively. Since the Naviglio extract presented a higher antimicrobial activity with a lower MIC and MBC value, this extract was chosen to make the nanoemulsions.

### 3.3. Brine Shrimp Bioassay

All extracts proved to be non-toxic in the same concentrations of MIC and MBC evaluated. It is important to mention that, for the Naviglio extract, the PROBIT method was analyzed, in which the average lethal dose $LD_{50}$ was 58 mg/mL.

### 3.4. Nanoemulsion Characterization

During ultrasonic nanoemulsification, factors such as emulsion formulation, emulsification conditions, and the amount of energy applied affect nanoemulsion characteristics [30]. In order to study the influence of these factors on the droplet size and distribution of the nanoemulsions, a factorial design was carried out where the concentration of the extract and the ultrasound amplitude were taken as factors, and the response variables were the droplet size and polydispersity index (PDI). The statistical analysis showed that all the variables and their interaction significantly affect the droplet size and PDI. The droplet size, PDI, and zeta potential of the nanoemulsions are depicted in Table 4. The NE2 and NE3 treatments presented a significantly smaller droplet size ($p < 0.05$) with $4.4 \pm 1.3$ and $5.8 \pm 1.2$ nm, respectively. In the case of nanoemulsions made with 5% of the extract, the ultrasound amplitude significantly influenced the droplet size. This agrees with what was described by Carpenter and Saharan [52], who observed that, as the amplitude increased, the droplet size in mustard oil nanoemulsions decreased because, at a higher amplitude, more energy is dissipated in the system, which produces higher cavitational collapse pressure, causing the disruption of large droplets into smaller ones. Similarly, Lago et al. [53] reported smaller particle sizes at a higher amplitude in *Pereskia aculeata* nanoemulsions; hence, inputting a large amount of mechanical energy is essential for producing smaller droplets.

**Table 4.** Droplet size, polydispersity index, and zeta potential of *Plectranthus hadiensis* var. *tomentosus* nanoemulsions.

| Treatment | Extract Concentration (%) | Amplitude (%) | Droplet Size (nm) [1,2] | Polydispersity Index [1,2] | Zeta Potential (mV) [1,2] |
|---|---|---|---|---|---|
| NE1 | 5 | 40 | $13.5 \pm 3.7$ [a] | $0.85 \pm 0.08$ [a] | $0.03 \pm 0.15$ [a] |
| NE2 | 5 | 50 | $4.4 \pm 1.3$ [b] | $0.48 \pm 0.03$ [b] | $-0.08 \pm 0.20$ [a] |
| NE3 | 10 | 40 | $5.8 \pm 1.2$ [bc] | $0.51 \pm 0.03$ [b] | $0.04 \pm 0.20$ [a] |
| NE4 | 10 | 50 | $7.7 \pm 0.6$ [c] | $0.59 \pm 0.07$ [c] | $0.06 \pm 0.17$ [a] |

[1] Values represent the average $\pm$ standard deviation of six measurements. [2] Within columns, values with different letters differ significantly ($p < 0.05$).

In addition to the ultrasound amplitude, the extract concentration significantly affects the droplet size in nanoemulsions with an amplitude of 50%. An increase in the droplet size was observed as the extract concentration increased. This was also reported by Carpenter and Saharan [52] and Sharma, Kaur, and Khatkar [54], where the increase in oil concentration produced nanoemulsions with bigger droplet sizes. According to Lago et al. [53], with the increase in oil concentration, the emulsifier in the medium is not enough to coat all the surfaces of the droplets, favoring aggregation and coalescence.

The polydispersity index indicates the droplet size uniformity within a nanoemulsion. Generally, PDI values lower than 0.10 are considered highly monodispersed, values of 0.1–0.4 are categorized as moderately polydispersed, and values above 0.40 are classified as highly polydispersed [54]. The PDI value of the nanoemulsions in this study ranged from 0.48 to 0.85, showing a polydisperse nature. As with the droplet size, the extract

concentration and ultrasound amplitude significantly affected the PDI. In the NE2 and NE4 treatments, an increase in the oil concentration increased the PDI value due to the exposed surface of oil droplets derived from the insufficient amount of emulsifier, causing the droplets' coalescence, leading to an increased PDI. In addition, in the case of the NE1 and NE2 nanoemulsions, an increase in the ultrasound amplitude causes a decrease in the PDI due to the input of higher energy that promotes the breakdown of the droplets into a homogeneous size distribution.

Regarding the zeta potential of the nanoemulsions, there was no significant difference ($p > 0.05$) between the treatments, presenting values close to zero, ranging from $-0.08$ to $0.06$ mV. This is due to the type of emulsifier used. Tween 40 is a non-ionic emulsifier, so it does not present a charge but prevents the aggregation of oil droplets through steric hindrance by generating a barrier with its molecular groups between the oil droplets and the continuous phase [32,55].

*3.5. Storage Stability Study*

The stability of the nanoemulsions was evaluated during 30 days of storage at room temperature, and changes in the droplet size and PDI were evaluated every week. None of the nanoemulsions presented phase separation and had droplet sizes below 200 nm, so they were all considered stable. According to Chu et al. [56], nanoemulsions with a droplet size below 70 nm have better stability against phase separation due to the droplets' Brownian motion and weaker colloidal interactions between the droplets. The changes in droplet size and PDI of the nanoemulsions during storage are shown in Figure 2. Nanoemulsions NE3 and NE4 had a significantly increased droplet size and PDI after storage. The increase in droplet size depends mainly on the fact that nanoemulsions tend to decrease their specific surface area and free energy in thermodynamically unstable systems, which leads to coalescence, flocculation, and Ostwald ripening [56]. In the case of nanoemulsion NE1, contrary to what was observed in the other nanoemulsions, it presented a decrease in droplet size and PDI after one week. Then, it remained constant throughout the following weeks. The decrease in droplet size can be due to the reconstruction of the nanoemulsion in order to reach a stable equilibrium state [57]. In contrast, nanoemulsion NE2 did not present a significant change ($p > 0.05$) in droplet size and PDI after 30 days, indicating greater stability. According to Lago et al. [53], a nanoemulsion is considered stable if it has a similar droplet size during storage. At the same time, instability can be confirmed if there is a significant variation in size during storage.

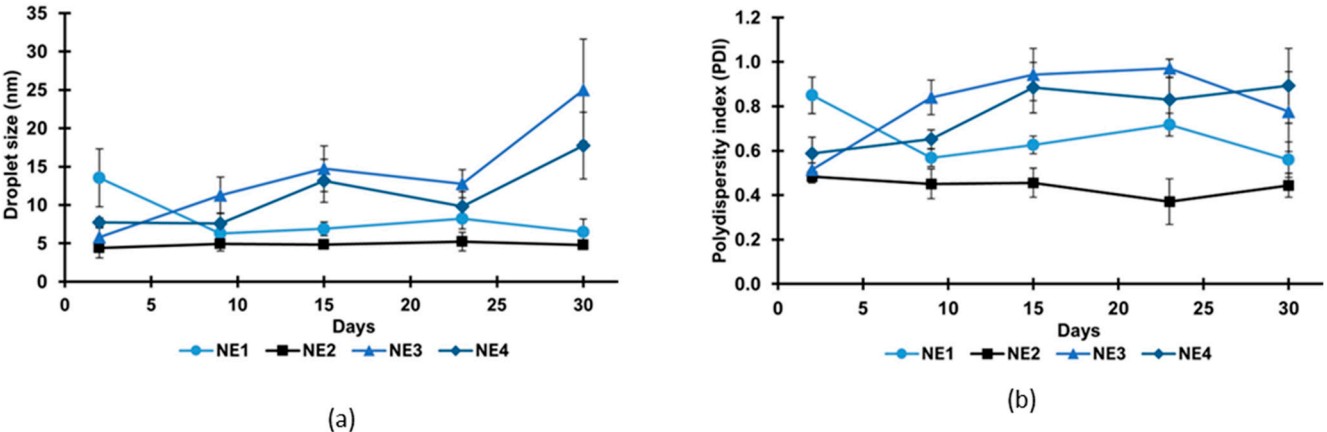

**Figure 2.** Changes in the droplet size (**a**) and PDI (**b**) of *Plectranthus hadiensis* var. *tomentosus* nanoemulsions during 30 days of storage.

In addition to analyzing the changes in droplet size and PDI of the nanoemulsions during storage, the total phenolic content and antioxidant capacity were also evaluated. The total phenolic content and antioxidant capacity of fresh nanoemulsions after 30 days of

storage are shown in Table 5. Regarding the total phenolic content, NE3 and NE4 significantly decreased ($p < 0.05$) in these compounds, while NE2 did not present a significant change in polyphenol content. This agrees with what was observed for the droplet size and PDI, which indicates the greater degree of instability in NE3 and NE4 that caused the degradation and loss of phenolic compounds. Regarding the antioxidant capacity, all nanoemulsions significantly increased ($p < 0.05$) in their activity after storage. Similar behavior was described by Bazana et al. [58] in nanoemulsions of *Physalis peruviana* extract stored at room temperature and protected from light, where they concluded that nanoencapsulation can protect the bioactive compounds in the extract against degradation, which favors a prolonged shelf-life. According to Liu et al. [57], the antioxidant capacity of the nanoemulsion does not decrease after storage due to the interfacial layer that provides a protective effect by reducing the diffusion of oxygen, free radicals, and pro-oxidants at the oil–water interface. The increase in antioxidant activity after storage is related to the release profile shown in Figure 3, but the total phenolic content and antioxidant activity decreased compared with the crude extract. In this regard, it could be assumed that, since the continuous phase was water, some hydrophilic compounds were not encapsulated and, hence, were lost during the process due to environmental factors [59]. It has been proved that the droplet size does not affect the antioxidant activity, but it keeps the nanoemulsion stable [60]. Thus, since the NE2 nanoemulsion did not present significant changes in droplet size, PDI, and total phenolic content during storage, this treatment was chosen for subsequent tests.

**Table 5.** Total phenolic content and antioxidant capacity of *Plectranthus hadiensis* var. *tomentosus* nanoemulsions.

| Treatment | Total Polyphenols (mg GAE/g) [1] | | Antioxidant Capacity (mM TE/g) | |
|---|---|---|---|---|
| | Day 1 | Day 30 | Day 1 | Day 30 |
| NE1 | 28 ± 0.9 [a] | 31 ± 1.1 [b] | 62 ± 2.7 [a] | 79 ± 15 [b] |
| NE2 | 27 ± 1.7 [a] | 27 ± 1.8 [a] | 60 ± 2.0 [a] | 74 ± 3.7 [b] |
| NE3 | 30 ± 1.6 [a] | 22 ± 0.5 [b] | 72 ± 5.7 [a] | 94 ± 5.9 [b] |
| NE4 | 27 ± 1.0 [a] | 22 ± 0.4 [b] | 67 ± 3.8 [a] | 94 ± 4.1 [b] |

[1] Values represent the average ± standard deviation of six measurements. Within rows, values with different letters (a, b) are significantly different ($p < 0.05$).

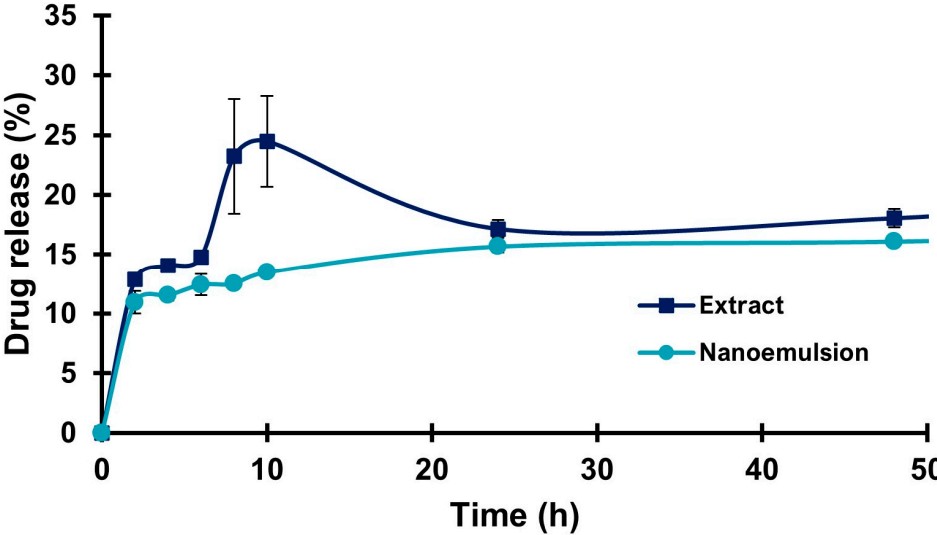

**Figure 3.** Percentage of drug release of the free extract and nanoemulsion.

### 3.6. Encapsulation Efficiency (EE)

The encapsulation efficiency for the most stable nanoemulsion was $95 \pm 0.8\%$, which agrees with what was reported by other authors. Jan et al. [30] obtained an EE of $91 \pm 0.4\%$ for the vitamin D3 nanoemulsion, while the eucalyptus essential oil nanoemulsion reported by Song et al. [61] had an EE of $97 \pm 1.2\%$. A curcumin nanoemulsion's highest EE ($94 \pm 1\%$) was obtained in a nanoemulsion with 5% oil, 20 min of ultrasound treatment, and 1.5% emulsifier [55]. According to Song et al. [61], high-frequency ultrasound provides high values of EE because it supplies enough energy to generate intense disruptive forces and minimize the droplet size of nanoemulsions, which helps to trap the bioactive compound within the encapsulant.

### 3.7. Drug Release Study

Figure 3 shows the in vitro release of the extract and the PHT nanoemulsion, where the extract presented a maximum release of $24.5 \pm 3.8\%$ after 10 h, followed by a decrease at 24 h, possibly due to a degradation of the released compounds. On the other hand, the nanoemulsion presented a release percentage of $13.5 \pm 0.2\%$ after 10 h and $16.2 \pm 0.1\%$ after 48 h, showing a more controlled release than the extract. Similar behavior was reported by Dávila-Rodríguez et al. [62] in curcumin nanoparticles, where free curcumin presented a release of 80% over 20 h while the nanoparticles presented only 40% over the same amount of time. The release profile of the nanoparticles was prolonged and constant compared with free curcumin. Similarly, Imam et al. [63] observed a faster release of pure naringin than the nanoencapsulated compound after 12 h (98.3% and 71.5%, respectively). Nanoencapsulation of bioactive compounds promotes a controlled release and increases their bioavailability [1,17] because the main factor that controls the release behavior is the interaction between the carrier matrix and the medium of liberation. Furthermore, this can also be influenced by the interaction between the bioactive compound and the other components of the nanoemulsion [31].

### 3.8. Phenolic Compound Quantification and Antioxidant Activity of the Nanoemulsion

The total phenolic content of the extract and nanoemulsion of PHT is shown in Figure 4. The nanoemulsion exhibited a significantly higher ($p < 0.05$) polyphenol content ($73 \pm 1.6$ mg GAE/g) than the extract ($23 \pm 1.6$ mg GAE/g). Badr, Badawy, and Taktak [64] obtained a higher polyphenol content in lavender oil nanoemulsions than in the free oil due to the improved availability of the phenolic content in the nanoemulsions. Similar results were observed in this study, where the concentration of polyphenols in the nanoemulsion is three times higher than that of the extract, which is attributed to an improvement in the availability and solubility of the phenolic compounds present in the extract due to the emulsifier used. In addition, reducing the particle size increases the surface contact area of the polyphenols during the quantification reaction.

The antioxidant capacity of the extract and nanoemulsion of PHT can be seen in Figure 5. In the DPPH test, the extract showed an activity of $194 \pm 6.4$ mM TE/g and the nanoemulsion showed an activity of $259 \pm 3.8$ mM TE/g, while in the FRAP method the antioxidant activity was $169 \pm 27$ mM TE/g and $379 \pm 11$ mM TE/g for the extract and the nanoemulsion, respectively. In both tests, the antioxidant capacity of the nanoemulsion was significantly higher ($p < 0.05$) than that of the unencapsulated extract, which agrees with previous reports. Doost et al. [65] evaluated the antioxidant capacity of thymol nanoemulsions by DPPH and FRAP methods and reported that nanoemulsions had a higher antioxidant capacity than free thymol. In the case of citrus essential oil nanoemulsions, the scavenging activity of the nanoemulsion (72.4%) was significantly higher than that of the pure essential oil (44.3%) [66]. Araujo et al. [67] and Seibert et al. [68] also reported that clove oil and propolis nanoemulsions showed a higher antioxidant capacity than the free extracts. The increase in the antioxidant capacity of the nanoemulsion compared with the extract can be due to different reasons. According to Doost et al. [65], the reduced droplet size of the nanoemulsions provides a greater specific surface area and, thereby, more active

sites exposed to free radicals. In addition, nanoemulsions show better solubility in aqueous solutions than free oils, which favors a rapid release of their active components in the solution, thus neutralizing free radicals more efficiently and increasing their antioxidant capacity [66,69].

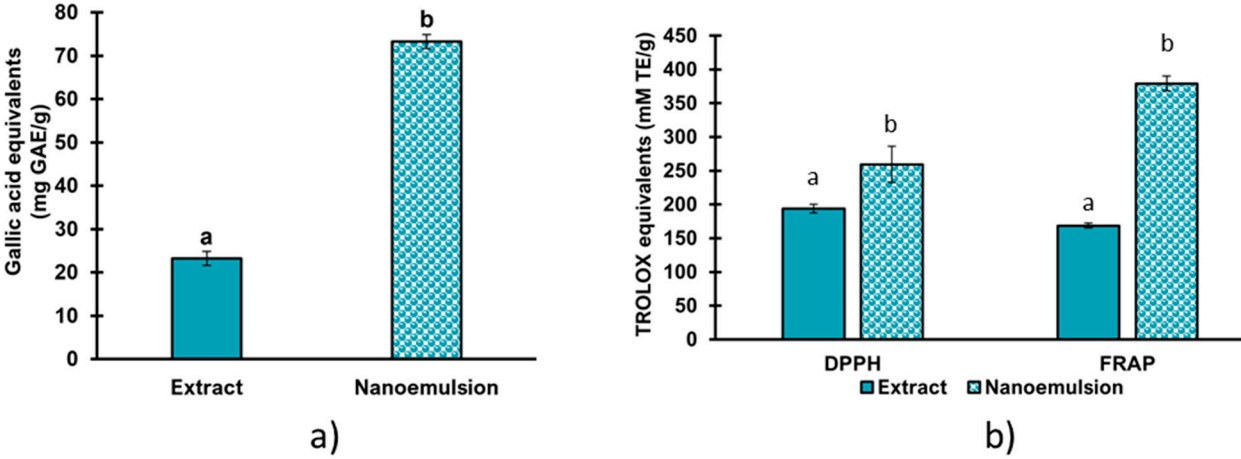

**Figure 4.** (**a**) Total polyphenol content of *Plectranthus hadiensis* var. *tomentosus* extract and nanoemulsion and (**b**) antioxidant capacity of *Plectranthus hadiensis* var. *tomentosus* extract and nanoemulsion. Values with different letters (a, b) are significantly different ($p < 0.05$).

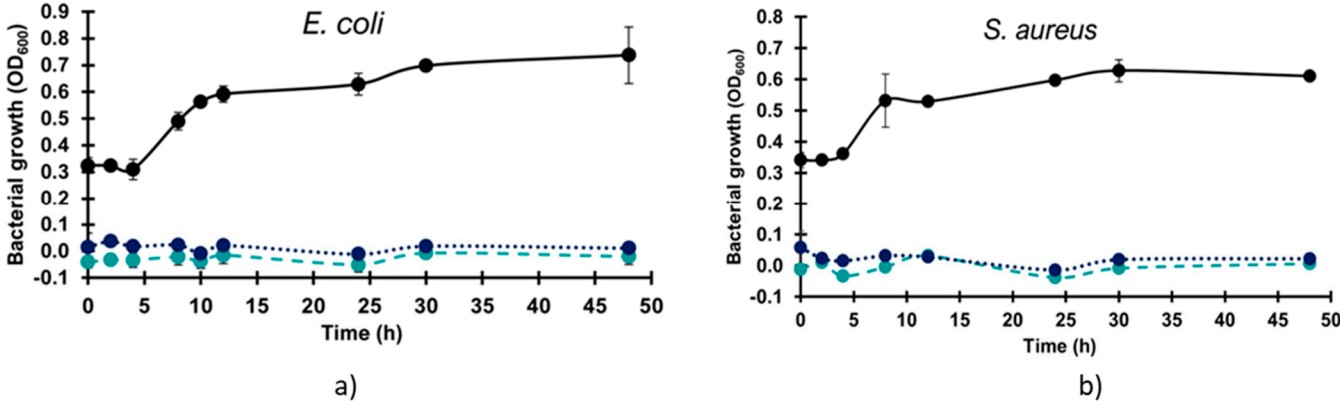

**Figure 5.** Time–kill kinetic curves of the free and encapsulated extract. Microbial survival of *Escherichia coli* (**a**) and *S. aureus* (**b**). Control (solid line), free extract (dotted line), and nanoemulsion (dashed line).

### 3.9. Antimicrobial Activity of the Nanoemulsion

The minimum inhibitory concentration and minimum bactericidal concentration of the PHT extract and nanoemulsion are shown in Table 6. In the case of the extract, the MIC was not detected for any of the four bacteria at the tested concentration, so it was reported as >25 mg/mL. The MIC of the nanoemulsion was 25 mg/mL, so the encapsulation was able to reduce the MIC of the free extract, which agrees with previous reports. Badr, Badawy, and Taktak [64] observed that lavender oil nanoemulsions exhibited a lower MIC than the free oil against *Salmonella* Typhimurium and *Staphylococcus aureus*. Qi et al. [33] also reported that D-limonene nanoemulsions reduced the MIC of the pure compound by 4–5 fold for *Escherichia coli* and *S. aureus*. Additionally, Kang et al. [70] observed a significant increase in the antimicrobial activity of citrus essential oil after encapsulation since nanoemulsions reduced the MIC value 43 times for *S. aureus* and 2.7 times for *E. coli* compared with the free oil. The increase in antimicrobial activity can be attributed to the fact that nanoemulsions reduce the hydrophobicity of bioactive compounds, resulting in a uniform distribution that allows molecules to have more contact

with the cell membrane surface of bacteria [65,71]. In addition, the reduced particle size allows the nanoencapsulated active compounds to penetrate the phospholipid bilayer of bacteria more easily because nanodroplets can increase the passive transport of the cell, overcome the low permeability of the bacterial membranes, and enhance the interaction with the intracellular transport, allowing membrane disruption and leakage of cellular components [57,68].

**Table 6.** Minimum inhibitory concentration (MIC) and minimum bactericidal concentration (MBC) of the extract and nanoemulsion of *Plectranthus hadiensis* var. *tomentosus*.

| Bacteria | MIC (mg/mL) [1] | | MBC (mg/mL) | |
|---|---|---|---|---|
| | Extract | Nanoemulsion | Extract | Nanoemulsion |
| *Staphylococcus aureus* | >25 | 25 | n.d. [2] | >25 |
| *Listeria monocytogenes* | >25 | 25 | n.d. | 25 |
| *Salmonella enterica* | >25 | 25 | n.d. | 25 |
| *Escherichia coli* | >25 | 25 | n.d. | >25 |

[1] Values represent the average of three measurements. [2] n.d. = not determined.

Regarding the MBC, for *S. aureus* and *E. coli*, the nanoemulsion showed an MBC of >25 mg/mL, while for *Listeria monocytogenes* and *Salmonella enterica* the MBC was 25 mg/mL. These results are in accordance with Özogul, El Abed, and Özogul [72], who obtained an MBC of 25 mg/mL for *Salmonella* Paratyphi A and >25 mg/mL for *S. aureus* in laurel essential oil nanoemulsions. Given that the MIC and MBC values for *L. monocytogenes* and *Salmonella* were the same, it is possible to consider that the nanoemulsion showed a bactericidal effect, while for the case of *E. coli* and *S. aureus* it exhibited a bacteriostatic effect because the MBC was higher than the MIC.

*3.10. Time–Kill Kinetics Assay*

In order to evaluate the effect of the nanoemulsion and the PHT extract on the growth of *Escherichia coli* and *Staphylococcus aureus*, the optical density at 600 nm was measured at different incubation times. Figure 5a,b show the effect on the growth of *E. coli* and *S. aureus*, respectively. In both cases, the control without the nanoemulsion or extract entered the logarithmic phase after 4 h and presented a maximum absorbance after 48 h of 0.74 and 0.61 for *E. coli* and *S. aureus*, respectively. On the other hand, the bacteria treated with the nanoemulsion or extract at the MIC did not show an increase in absorbance after 48 h, which means that both the nanoemulsion and the extract were able to inhibit the growth of the two microorganisms significantly. These results coincide with those reported by Qi et al. [33], who determined the effect on the growth of *E. coli* and *S. aureus* treated with the D-limonene nanoemulsion at the MIC and observed that there was no increase in absorbance after 8 h of incubation, so they concluded that the nanoemulsion significantly inhibited the growth of the bacteria because the encapsulation allowed for a controlled release of the bioactive compound, thus prolonging the inactivation of the microorganism. Similarly, Liu et al. [57] determined the effect on the growth of methicillin-resistant *Staphylococcus aureus* (MRSA) treated with the garlic essential oil nanoemulsion at different concentrations. They observed that the MRSA treated with the nanoemulsion at the MIC did not show an increase in absorbance after 24 h.

**4. Conclusions**

The Naviglio method allows us to obtain extracts with antimicrobial activity at lower concentrations compared with other traditional methods. Through the ultrasound technique and using Tween 40 as a surfactant, it is possible to obtain nanoemulsions with droplet sizes smaller than 100 nm and narrow size distributions that are compatible with

PHT extracts, which are favored by an enhancement in their antioxidant and antimicrobial activity against pathogens of interest found in food. These nanosystems are excellent candidates for application in different compatible food matrices, such as the meat industry or the export of horticultural products, due to the stability, enhancement in antioxidant activity, and antimicrobial power demonstrated against both Gram-positive and Gram-negative pathogens.

**Author Contributions:** Conceptualization, A.C.-G. and J.M.S.-J.; methodology, L.C.V.-H., J.C.S.-N. and C.A.V.-C.; software, L.C.V.-H., A.C.-G. and J.M.S.-J.; formal analysis, L.C.V.-H., J.M.S.-J. and A.C.-G.; investigation, L.C.V.-H., J.M.S.-J. and C.A.V.-C.; validation, A.H.M.-P., J.C.S.-N. and C.A.V.-C.; manuscript review and editing, J.M.S.-J., L.C.V.-H. and A.C.-G.; visualization, J.M.S.-J. and A.C.-G.; resources, A.C.-G., J.M.S.-J., A.H.M.-P. and J.C.S.-N. All authors have read and agreed to the published version of the manuscript.

**Funding:** Thanks to CONHAC$_\gamma$T for the scholarship to Vega-Hernández (Master in Science grant 792661).

**Data Availability Statement:** The data presented in this study are available upon request from the corresponding author.

**Acknowledgments:** Thanks to Elisa Cabrera Díaz and María Esther Macías Rodríguez for the strains provided for this project.

**Conflicts of Interest:** The authors declare no conflict of interest.

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
