# Peer review of "Improving Foodborne Pathogen Control Using Green Nanosized Emulsions of Plectranthus hadiensis Phytochemicals"

_colloids, doi:10.3390/colloids8010003_

Round 1

Reviewer 1 Report

Comments and Suggestions for Authors

The study: "Improving Foodborne Pathogens Control with Green Nanosized Emulsions of Plectranthus hadiensis Phytoconstituents" is well designed and well conducted, statistical methods and analysis are correct. Discussion and conclusion are adequate.

Author Response

Thank you very much for your comments on our work.

Reviewer 2 Report

Comments and Suggestions for Authors

This manuscript indicates " Improving Foodborne Pathogens Control with Green Nanosized Emulsions of Plectranthus hadiensis Phytoconstituents".  Some of the revisions and comments are summarized below. Others are given in the manuscript in the attached file. 

-The title should be clarified. 

- Citation of references should be done according to the " Introduction to the Authors". 

-In Material and Methods, some experiments are missing. Cytotoxicity and total flavonoid content of extracts and nanoemulsions should be discussed. 

-FTIR results should be discussed in more detail by comparing literature. 

- In antimicrobial analysis, results are confusing. MIC and MBC values for extracts ( naviglio) in Table 3 and Table 6 are different. Why? You stated that Naviglio extract was selected for nanoemulsion production.  If you used the same extract, the values should be the same.  

- You selected Naviglio extract for further experiments. there are no statistical differences between the polyphenol content of extracts obtained by ultrasound and Naviglio. In addition, you obtained the highest antioxidant capacity by using ultrasound extraction. You should discuss. 

Line 307-314: Differences in phenolics for extraction methods were discussed for other references.  But in this study, it is missing. The phenolic profile of extracts and changes in nanoemulsions should be investigated.  

- Most statistical results are shown in tables and figures. but some of them are missing. And subscripts should be used to explain the meanings of letters. 

-Standard deviations for some points are very high (Fig. 3, 5). 

Comments on the Quality of English Language

Extensive editing of English language is required. There are  grammatical errors and unclear sentences in the manuscript

Author Response

We appreciate the comments, and below, we break down what is requested point by point. This manuscript indicates " Improving Foodborne Pathogens Control with Green Nanosized Emulsions of Plectranthus hadiensis Phytoconstituents". Some of the revisions and comments are summarized below. Others are given in the manuscript in the attached file.

All responses written as “done” are highlighted in the manuscript. Also, the comments inserted in the PDF document made by the reviewer were highlighted.

-The title should be clarified.

Done. We suggest changing the title to “Improving Foodborne Pathogens Control using Green Nanosized Emulsions of Plectranthus hadiensis Phytochemicals.”

- Citation of references should be done according to the " Introduction to the Authors."

The citations highlighted were corrected.

-In Material and Methods, some experiments are missing. Cytotoxicity and total flavonoid content of extracts and nanoemulsions should be discussed.

Done. Total flavonoid content and Artemia salina bioassay were included.

-FTIR results should be discussed in more detail by comparing literature.

Done. In the line, you will find the expanded discussion in the highlighted text.

- In antimicrobial analysis, results are confusing. MIC and MBC values for extracts (Naviglio) in Table 3 and Table 6 are different. Why? You stated that Naviglio extract was selected for nanoemulsion production. If you used the same extract, the values should be the same.

In line 271, we describe that Tween was added to the extract to avoid any false positive of the surfactant and to be comparable with the composition of the nanoemulsion. Thus, the 100 uL of only extract in the antimicrobial test is different from the presented activity of the extract in the total nanoemulsion composition (without encapsulation).

Comment in PDF file (line 257): Only those cultures were selected because they are the most representative since they have resistance mechanisms that make them resistant to multiple antimicrobials, generating therapeutic challenges. In addition, they represent both the group of gram-positive and gram-negative bacteria.

- You selected Naviglio extract for further experiments. there are no statistical differences between the polyphenol content of extracts obtained by ultrasound and Naviglio. In addition, you obtained the highest antioxidant capacity by using ultrasound extraction. You should discuss.

Our application of interest is aimed at food safety, so we selected the technique that presented the most significant antimicrobial activity. Furthermore, high-value antioxidant activity is not always the best option since it can offer counterproductive effects and even be pro-tumor. We decided to use Naviglio due to the antimicrobial capacity presented.

Line 307-314: Differences in phenolics for extraction methods were discussed for other references. But in this study, it is missing. The phenolic profile of extracts and changes in nanoemulsions should be investigated.

Polyphenols, flavonoids and total antioxidant activity for each extraction method can be found in table 2, and for antioxidant activity of nanoemulsions with different concentrations of extract are mentioned in table 5.

- Most statistical results are shown in tables and figures. but some of them are missing. And subscripts should be used to explain the meanings of letters.

In antioxidant activity figure, letters indicating statistical differences were added.

-Standard deviations for some points are very high (Fig. 3, 5).

This resulted from three replications, so we must run other experiments. Furthermore, it is only in some two points of all those presented in the curves, and, it should be noted that when working with extracts, it is challenging to obtain slight standard deviations.

-In the case of the extract, the MIC was not detected for any of the four bacteria at the tested concentration, so it was reported as >25 mg/ml.

It should be noted that in this test, the antibacterial activity of the extract was studied by mixing the other components of the nanoemulsion, but they weren’t submitted to ultrasound. This was applied to determine the effect of the ultrasound technique.

Reviewer 3 Report

Comments and Suggestions for Authors

The authors reported "Improving Foodborne Pathogens Control with Green Na-nosized Emulsions of Plectranthus hadiensis Phytoconstituents". The manuscript is well-written and presented. However, the authors should consider the following comments for the better stand. 

Line number 24: The abbreviation of MIC and MBC needs to be incorporated 

Line numbers 152-153: The percentage of methanol could be added 

Line numbers 163-165: Authors could include the r square value and the equation of a straight line of the standard curve when calculating ferric-reducting activity. 

Line number 291-294: Authors can expand the section with supporting citations. 

Line number 449-450: Supporting sentence and citations are needed to support this sentence: "Since NE2 448 nanoemulsion did not present significant changes in droplet size, PDI, and total phenolic 449 content during storage, this treatment was chosen for subsequent tests."

Line number 569-572: Authors could revise the sentence and expand the section. 

Author Response

We appreciate your comments and improvements to our work. Below, you will find details about your suggestions.

The authors reported "Improving Foodborne Pathogens Control with Green Na-nosized Emulsions of Plectranthus hadiensis Phytoconstituents". The manuscript is well-written and presented. However, the authors should consider the following comments for the better stand.

Line number 24: The abbreviation of MIC and MBC needs to be incorporated
Done.
Line numbers 152-153: The percentage of methanol could be added
Done.
Line numbers 163-165: Authors could include the r square value and the equation of a straight line of the standard curve when calculating ferric-reducting activity.
Done.
We added the respective standard curves with the determination coefficient for all phytochemical tests.
Line number 291-294: Authors can expand the section with supporting citations.
A reference of the effect of Soxhlet extraction in antioxidant activity was included.
Line number 449-450: Supporting sentence and citations are needed to support this sentence: "Since NE2 448 nanoemulsion did not present significant changes in droplet size, PDI, and total phenolic 449 content during storage, this treatment was chosen for subsequent tests."
A reference concerning this point was added.
Line number 569-572: Authors could revise the sentence and expand the section.
It was extended in accordance with the results.

Round 2

Reviewer 2 Report

Comments and Suggestions for Authors

Some of the missing experiments are included in the manuscript. Discussions are improved.  However, the answer to the comment "Line 307-314: Differences in phenolics for extraction methods were discussed for other references. But in this study, it is missing. The phenolic profile of extracts and changes in nanoemulsions should be investigated." is not satisfactory.   If the phenolic profile of the extract could not be included in to the manuscript. The related part should be revised due to it may cause confusion. 

Comments on the Quality of English Language

Language should be checked 

Author Response

Thank you for the observation. We include two references about the phenolic profile of P. hadiensis since in our study we didn’t performed an HPLC analysis. In the manuscript you can find the information highlighted.
